# Dysregulation of SIRT3 SUMOylation Confers AML Chemoresistance via Controlling HES1-Dependent Fatty Acid Oxidation

**DOI:** 10.3390/ijms23158282

**Published:** 2022-07-27

**Authors:** Yirong Zhang, Yajie Shen, Weiqing Wei, Wenhan Wang, Daiji Jiang, Yizhuo Ren, Zijing Peng, Qiuju Fan, Jinke Cheng, Jiao Ma

**Affiliations:** 1Department of Biochemistry and Molecular Cell Biology, Shanghai Jiao Tong University School of Medicine, Shanghai 200025, China; yirong@shutcm.edu.cn (Y.Z.); ashen9618@163.com (Y.S.); zodiac.d@sjtu.edu.cn (W.W.); wwhdou@sjtu.edu.cn (W.W.); captain_of_gangguo@sjtu.edu.cn (D.J.); 19341275@sjtu.edu.cn (Y.R.); pengzijing@sjtu.edu.cn (Z.P.); fanqiuju93@shsmu.edu.cn (Q.F.); jkcheng@shsmu.edu.cn (J.C.); 2Department of Biological Science, School of Life Science, Shaanxi Normal University, Xi’an 710119, China

**Keywords:** SIRT3 SUMOylation, AML chemoresistance, HES1, fatty acid oxidation

## Abstract

Sirtuin 3 (SIRT3) deacetylase is a key regulator for chemoresistance in acute myeloid leukemia (AML) cells due to its capability of modulating mitochondrial metabolism and reactive oxygen species (ROS). SIRT3 is de-SUMOylated by SUMO-specific peptidase 1 (SENP1), which enhances its deacetylase activity. Therefore, dysregulation of SIRT3 SUMOylation may lead to fortified chemoresistance in AML. Indeed, SIRT3 de-SUMOylation was induced by chemotherapeutic agents, which in turn, exacerbated resistance against chemotherapies in AML by activating SIRT3 via preventing its proteasome degradation. Furthermore, RNA-seq revealed that expression of a collection of genes was altered by SIRT3 de-SUMOylation including inhibition of transcription factor Hes Family BHLH Transcription Factor 1 (HES1), a downstream substrate of Notch1 signaling pathway, leading to increased fatty acids oxidation (FAO). Moreover, the SENP1 inhibitor momordin-Ic or HES1 overexpression synergized with cytarabine to eradicate AML cells in vitro and in xenograft mouse models. In summary, the current study revealed a novel role of SIRT3 SUMOylation in the regulation of chemoresistance in AML via HES1-dependent FAO and provided a rationale for SIRT3 SUMOylation and FAO targeted interventions to improve chemotherapies in AML.

## 1. Introduction

The Sirtuin family consists of seven family members (Sirtuin 1–7) that display diverse functions from regulation of chromatin structure to maintenance of mitochondrial homeostasis [1]. All sirtuins share a common nicotinamide adenine dinucleotide (NAD)+ domain but differ in their amino and catalytic domain, which determines their subcellular localizations [2]. Sirtuin 1 is primarily located in the cell nucleus but transmits from the cytosol and cell nucleus when cells are demanding energy and during the developmental stage [3]. Sirtuin 2 is primarily located in the cytoplasm, but can be found in the nucleus during the mitosis stage, Sirtuins 3, 4, and 5 are predominately located in mitochondria [4], Sirtuin 6 is located in cell nucleus [5] and Sirtuin 7 is located in cell nucleolus [6]. Sirtuin family proteins are known to participate in many physiological and pathological events including stress response [7], autophagy [8], tumorigenesis [9], longevity [10], and mitochondrial metabolism [11].

SIRT3 acts as a crucial scavenger for reactive oxygen species (ROS) by de-acetylating and activating anti-oxidant enzymes including SOD2 and IDH in mitochondria [12]. SIRT3 mediated reprogramming of cellular metabolism is critical for chemoresistance in a variety of tumors. For instance, increased SIRT3 expression activates SOD2 to reduce ROS generation leading to chemoresistance in colorectal cancer [13]. Similarly, augmented SIRT3 deacetylase activity mediated reprogramming of mitochondria oxidative phosphorylation confers AML chemoresistance via induction of SOD2 activity and inhibition of mitochondria ROS production in AML cell lines in response to ara-C [14]. However, data published in recent years support that SIRT3 can interact with non-mitochondrial proteins, in particularly transcription factors to regulate the expression of a variety of genes [15,16].

SUMOylation at the lysine residues may alter the protein–protein interaction, or exert a direct impact on the function of the substrate proteins including stability and localization by conformational changes [17]. Small ubiquitin-like modifiers (SUMO) can be removed from target proteins by SUMO-specific proteases (SENPs) via de-SUMOylation. A previous study showed that SIRT3 is a SUMOylated protein in mitochondria [18]. Interestingly, SENP1-mediated de-SUMOylation of SIRT3 occurs in response to fasting enhances SIRT3 deacetylase activity, and subsequently alters mitochondrial metabolism, such as oxidative phosphorylation and fatty acid oxidation (FAO) [18]. However, the molecular mechanism of SUMOylation-mediated SIRT3 deacetylase activity regulation remains yet to be explored.

## 2. Results

### 2.1. Chemotherapy Attenuates SIRT3 SUMOylation in AML

SUMOylation occurs at Lys288 of the human SIRT3 [18]. To determine whether SIRT3 SUMOylation also occurs in AML, the whole cell protein lysates were extracted from vector control, SIRT3, SIRT3K288R overexpressing MV4-11 cells and immunoprecipitated with SUMO1 antibody followed by immunoblotting with SIRT3 antibody. SIRT3 overexpressing MV4-11 cells showed a higher level of SUMOylated (Figure 1a and Appendix A), whereas SUMOylation of SIRT3K288R was inhibited (Figure 1a). To confirm that SIRT3K288R cannot be SUMOylated, myc-tagged SIRT3 or SIRT3K288R was transduced into MV4-11 cells. SUMOylation was only detected in myc-tagged SIRT3 overexpressing MV4-11 cells (Figure 1b). Consistent with the previous findings in hepatocellular carcinoma cells, these data demonstrated that SIRT3 is capable of being SUMOylated at Lys288 in AML cells.

SIRT3 inhibits the production of mitochondrial ROS via, at least in part, deacetylating metabolic enzymes in mitochondria, which contributes to AML chemoresistance [14]. SUMOylation also resulted in inhibition of SIRT3 deacetylase activity [18]. To determine whether SUMOylation of SIRT3 also participates in AML chemoresistance, SIRT3 SUMOylation was detected in AML cells upon treatment either with 2.5 μM Ara-C or 50 nM DNR at 48 h time points. As a result, the SIRT3 SUMOylation level was downregulated upon treatment (Figure 1c,d and Appendix A). This was confirmed in vivo by showing that SIRT3 SUMOylation was higher in Ara-C sensitive primary cells, but relatively low in resistant primary cells (see Appendix A). These results suggested that de-SUMOylation of SIRT3 is an important event in AML following chemotherapies.

### 2.2. De-SUMOylation Activates SIRT3 via Inhibition of Its Protein Degradation

To elucidate the mechanism of how SIRT3 deacetylase activity is regulated by de-SUMOylation, AML stable transfectants were treated with 300 μg/mL of CHX followed by determination of SIRT3 protein level at the indicated time points. As a result, the SIRT3 protein level was significantly decreased at 6 h in vector control overexpressing MV4-11 (Figure 2a top panel and Figure 2b) and MOLM-13 cells (Appendix A). However, SIRT3K288R remained barely changed within 9 h (Figure 2b middle panel, Figure 2c and Appendix A). To explore that SIRT3K288R possesses a longer half-life due to the loss of SUMOylation, momordin-Ic, a SENP1 inhibitor (Appendix A) was used to treat MV4-11/SIRT3K288R cells for 48 h. Percentages of the relative SIRT3 protein expressions were drastically dropped from about 100% to 12% at 9 h time points (Figure 2b bottom panel and Figure 2c), which was also observed in momordin-Ic treated MOLM-13/SIRT3K288R cells (Appendix A). Proteasome degradation of SIRT3 was then explored in MV4-11 cells transduced with Flag-tagged ubiquitin. Indeed, SIRT3 ubiquitination was significantly elevated upon exposure to momordin-Ic (Figure 2d left panel) or co-transduced with shSENP1 (Figure 2d right panel). Collectively, these data indicated that de-SUMOylation activates SIRT3 via inhibition of its protein degradation.

### 2.3. SIRT3 de-SUMOylation Confers AML Chemoresistance In Vitro and In Vivo

To explore the impact of SIRT3 SUMOylation on AML chemoresistance in vitro, lentivirus encoding vector control, SIRT3 or SIRT3K228R overexpressing MV4-11 cells were treated with the indicated doses of Ara-C for 48 h. MV4-11/SIRT3K288R cells displayed higher resistance to Ara-C as well as to DNR (Figure 3a,b). This was confirmed in vivo by demonstrating that SIRT3K288R xenografted mice were more resistant to Ara-C (Figure 3c). These data suggest that SIRT3 de-SUMOylation may contribute to a universal mechanism of drug resistance against a variety of chemotherapies in AML. Moreover, we established patient-derived xenograft (PDX) mouse models either from AML16 or AML18 [14] to evaluate their sensitivity to Ara-C. Upon treatment, about 25% of AML blasts (average) remained in AML18 xenografted mice, which exhibit a high level of acetylated SOD2 level and reflect low SIRT3 activity. In contrast, about 75% of AML blasts (average) remained in AML16 xenografted mice, which manifests a low level of acetylated SOD2 level and reflects high SIRT3 activity (Figure 3d). These data indicate that de-SUMOylation mediated activation of SIRT3 is a critical mechanism involved in the regulation of sensitivity to chemotherapeutic agents in AML.

### 2.4. SIRT3 SUMOylation Regulates Mitochondrial Biogenesis in AML

Since AML cells harbor a unique mitochondrial metabolism profile compared to solid tumors, the impact of SIRT3 SUMOylation on mitochondria biogenesis in AML was thus determined. The increase in NAD^+^/NADPH but the decrease in GSH/GSSG ratio was less pronounced in SIRT3K288R transduced AML cells (Figure 4a,b). Moreover, overexpression of SIRT3K288R upregulated OCR but inhibited ECAR in AML cells treated either with or without Ara-C (Figure 4c,d). Furthermore, SIRT3K288R stabilized Ara-C induced MMP (Figure 4e). Collectively, these data indicate that SIRT3 SUMOylation is involved in the regulation of mitochondrial biogenesis in response to extracellular stress.

### 2.5. SIRT3 de-SUMOylation Confers AML Chemoresistance via Down-Regulating HES1 Dependent FAO

To investigate the molecular mechanisms involved in SIRT3 de-SUMOylation induced AML chemoresistance. Lentiviral encoding vector control, SIRT3 or SIRT3K288R transduced MV4-11 cells were subjected to RNA-seq analysis (GSE179617 and see data availability). Thirty genes were found to be simultaneously downregulated (Figure 5a) and ten were validated by qRT-PCR (See Appendix A), among which HES1 was one of the most significantly downregulated genes (100-fold reduction relative to vector control) in SIRT3K288R transduced cells (Figure 5b). It has been reported that inhibition of HES1 enhances fatty acids oxidation (FAO) [19], presumably via crosstalk between Notch and PI3K/AKT signaling pathways [20]. Indeed, expressions of Notch1, MOLM1, BPUSH, HES1 proteins, phosphorylation of PI3K, ATK, and p38 proteins were significantly downregulated in SIRT3K288R overexpressing cells (Figure 5c and Appendix A). To determine whether SUMOylation of SIRT3 directly modulates HES1, the HES1 mRNA level was then explored in AML cells. As a result, HES1 mRNA was dramatically increased in 3-TYP or momordin-Ic treated cells compared to vehicle control (Figure 5d). To further confirm if SIRT3 SUMOylation downregulates HES1 via enhancing FAO, we explored the FAO fluxes in AML cells transduced either with empty vector, SIRT3 WT, or SIRT3 K228R lentiviral plasmids. As expected, de-SUMOylation of SIRT3 showed much stronger activity in inducing FAO, which was attenuated by HES1 overexpression (Figure 5e). Moreover, overexpression of HES1 counteracted SIRT3K288R-induced AML chemoresistance by decreasing AML cells survival (Figure 5f). To further explore whether SIRT3K288R induced AML chemoresistance via inhibition of FAO, we treated SIRT3K288R lentiviral plasmid overexpressing MV4-11 cells with either Ara-C, etomoxir, an FAO inhibitor, alone, or both for 48 h. Indeed, FAO fluxes were dramatically decreased upon etomoxir treatment compared to vehicle control (Figure 5g). Moreover, the FAO inhibitor etomoxir rescued sensitivity to chemotherapy in AML cells (Figure 5h). Taken together, these results indicate that SIRT3 de-SUMOylation mediated AML chemoresistance may be via HES1-dependent FAO in AML.

### 2.6. Inhibition of SIRT3 de-SUMOylation Synergizes with Ara-C in AML In Vitro

Enhanced chemoresistance by SIRT3 de-SUMOylation provided the rationale for inhibiting the SIRT3 de-SUMOylation process to improve the anti-leukemic efficacy of chemotherapeutic agents. Not surprisingly, SIRT3 SUMOylation and SUMO1 protein levels were significantly increased but SIRT3 activity was inhibited in MV4-11 cells treated with momordin-Ic. (Figure 6a). We then explored whether AML cells respond to momordin-Ic in a SIRT3 SUMOylation-dependent manner. Figure 6b showed that AML cells expressing SUMOylation deficient SIRT3K288R were more resistant to momordin-Ic than their vector control counterparts. SIRT3 transduced MV4-11 cells, which are relatively resistant to Ara-C and capable of responding to momordin-Ic, were treated with 10, 15, 20, 25 μM of momordin-Ic or 2.5, 5, 10, 15 μM of Ara-C alone or in combination to determine if momordin-Ic synergizes with chemotherapy in AML cells. As a result, synergism was observed in cells treated with a constant ratio of Ara-C: momordin-Ic (1:5) (Figure 6c). This was further confirmed by flow cytometry analysis of induced mitochondria ROS and cleaved caspase 3 activity (Figure 6d,e). Collectively, these data suggest that synergism of momordin-Ic with a chemotherapeutic agent can eradicate more AML bulks.

### 2.7. Combination Therapy Targeting SIRT3 SUMOylation Pathways

To explore if SIRT3 SUMOylation-related pathways including upstream SENP1 and downstream HES1 can be targeted as potential therapeutic interventions against chemoresistance in AML, a combination of momordin-Ic, HES1 overexpression, and Ara-C was evaluated. Human leukemia xenografted mice were treated with Ara-C or/and momordin-Ic prior to sacrifice (Figure 7a). A combination of momordin-Ic and Ara-C eradicated a significant amount of AML blasts when compared to Ara-C or momordin-Ic alone (Figure 7b) and decreased the survival of mice (Figure 7c). To identify the role of HES1 in SIRT3 SUMOylation-mediated regulation of sensitivity to chemotherapies, SIRT3 transduced alone or SIRT3 and HES1 co-transduced MV4-11 cells were engrafted in NOD/SCID mice. Four weeks post-transplantation, mice were treated with Ara-C. As a result, SIRT3 and HES1 co-transduced xenotransplants were more sensitive to chemotherapy when compared to SIRT3 alone xenotransplants (Figure 7d). These data suggest that inhibition of SIRT3 SUMOylation may be a new approach to enhance the potency of chemotherapeutic agents for a better clinical outcome.

## 3. Discussion

One protein can either be modified by SUMOylation or by ubiquitination at the same time but with different consequences [21]. For instance, the SUMO E3 ligase PIAS3 induces polyubiquitination and SUMOylation of NR4A1, and mutation of NR4A1 SUMOylation enhanced NR4A1 stability [22]. Our study demonstrated that inhibition of SENP1 by momordin-Ic enhanced SIRT3 SUMOylation resulted in increased proteasome degradation. Collectively, these results indicate that there is a crosstalk between these two modifications in some cases, and, as an example, SENP1 mediated SIRT3 de-SUMOylation activates SIRT3 by preventing its proteasome degradation.

Dysregulation of SUMOylation has been implicated in many pathological diseases such as cancer [23,24]. Whereas SIRT3 SUMOylation deficiency overcomes high-fat diet (HFD) induced obesity in mice under fasting conditions has been documented by our previous study [18]. In the current work, we found that SIRT3 de-SUMOylation was induced by Ara-C, and its level may correlate with the sensitivity of primary AML cells in response to chemotherapy. This result indicates that SIRT3 SUMOylation may play a critical role in controlling AML chemo-sensitivity.

Chemoresistance is the biggest challenge to chemotherapies targeting AML. Therefore, a better understanding of the mechanisms of chemoresistance is crucial to improving the clinical outcome. Although our previous studies have suggested the deacetylase activity of SIRT3 in modulating mitochondria ROS and oxidative phosphorylation altered the chemo-sensitivity of AML to a chemotherapeutic agent, an increasing number of studies demonstrated a new role of SIRT3 in transcriptional regulation. For instance, SIRT3 deacetylases Foxo3a and promotes its entry into the cell nucleus by upregulating the expression of Foxo3a [15,25,26,27]. Our RNA-seq data revealed that the expression of HES1, a transcription factor, was significantly downregulated by SIRT3 alongside the inhibition of notch signaling proteins. Moreover, HES1-dependent FAO was enhanced by de-SUMOylation mediated activation of SIRT3.

Increasing reports demonstrated that FAO contributed to chemoresistance in gastric [28], breast [29], ovarian [30], and hepatocellular carcinomas [31]. Pharmacological inhibition of FAO sensitized leukemia cells to apoptosis [32], and synergetic of FAO inhibitors with Ara-C eradicated bone marrow resident chemoresistance AML cells [33]. In the current study, we demonstrated that de-SUMOylation of SIRT3 leads to increased FAO, which can be attenuated by both FAO inhibitor and HES1 overexpression, indicating that SIRT3 deacetylase activity may impact intracellular signal transduction such as downregulation of HES1, and thus enhance FAO. Inhibition of FAO by overexpressing HES1 alleviated SIRT3-induced chemoresistance in AML, suggesting that HES1 may be one of the key tumor suppressors outside the mitochondria targeted by SIRT3. However, molecular mechanisms that drive SIRT3 transcription regulation of HES1 still need to be addressed in the future.

Targeting SUMOylation to improve clinical outcomes was reported in a panel of tumor models including ovarian [34], lung [35], and melanoma [36]. Our data demonstrated the preliminary therapeutic value of utilizing the novel SIRT3 SUMOylation targeted intervention, such as inhibition of SENP1, to deactivate SIRT3 and re-sensitize AML cells. Furthermore, HES1-mediated FAO may also be a promising target to evaluate its therapeutic value in refractory/relapsed AML.

## 4. Materials and Methods

### 4.1. Drug Compounds and Antibodies

Ara-C and DNR were purchased from the Sigma-Aldrich (St. Louis, MI, USA). NAC, Momordin-Ic and 3-TYP were consumed from Selleckchem (Houston, TX, USA). All compounds, except for in vivo studies, were reconstituted in the DMSO, stored at 100 mM stock concentrations in −80 °C, and used at the indicated doses as suggested by the vendor. Flow cytometry antibodies, Alexa Fluor 647 Rabbit anti-Active caspase 3 and APC-H7 Mouse anti-Human CD45 were purchased from BD pharmingen (San Jose, CA, USA). PE/Cy5 anti-Mouse CD45 (clone 30-F11) was consumed from BioLegend (San Diego, CA, USA). Immunoblotting antibodies, SUMO1, SIRT3, SENP1, Notch Activated Targets Antibody Sampler Kit including Notch1 (FL), MAML1, BPUSH and HES1, p-PI3K p85, t-PI3K p85, p-p38, t-p38, p-AKT and t-AKT were purchased from Cell Signaling Technology (Danvers, MA, USA). Acetylated SOD2 and SOD2 antibodies were purchased from Abcam (London, UK). Tubulin and β-actin antibodies were purchased from Proteintech (Rosemont, IL, USA).

### 4.2. Cell Lines, Primary Cells and Culture Conditions

AML cell lines MV4-11 and MOLM-13 were cultured in Iscove’s Modified Dulbecco’s Medium (IMDM, Thermofisher Scientific, Waltham, MA, USA, Cat. No. 12440053) supplemented with 10–20% fetal bovine serum (FBS) (Thermofisher Scientific, Cat. No. 10099141C) and 100 g/mL penicillin/streptomycin (Thermofisher Scientific, Cat. No. 15140122). All primary cells were thawed and sub-cultured as previously described [14].

### 4.3. Lentiviral Plasmids Packaging

PCDH-vector control, SIRT3 WT, or SIRT3 K288R plasmids were co-transfected with two lentiviral packaging plasmids psPAX2 and pMD2G into 293T cells with lipofection (YEASEN 40802ES03) at ratio of 3:1:4, respectively. Viral supernatants were then collected at 48 and 72 h time points and enriched with addition of Lenti-X-Concentrator at ratio of 4:1. Viral pellets were obtained by centrifugating mixture at 1500g for 45 min at 4 °C prior to dissolved in serum and antibiotic-free IMDM at 1/100 of the original volume. Viral titer was then determined by qPCR using lentiviral concentration detection kit (Abm-LV900) and stored at −80 °C until further use.

### 4.4. Flow Cytometry Sorting of Stable Transfectants

Two million AML cells were resuspended in serum-free IMDM with addition of 6 μg/mL polybrene to induce lentiviral transduction before they were mixed either with 20 MOI of lentiviral encoding vector control, SIRT3 WT or SIRT3 K288R, respectively. Lentivirus was then attached to cells by centrifuging the culture plate at 1000× *g* for 1 h at 37 °C. Four hours post centrifugation process; extra fetal bovine serum was added onto cells and sub-cultured in an incubator. Cells were then suspended in fresh culture medium 24 h post lentiviral transduction prior to subject to GFP^+^/7AAD^−^ cell sorting using BD FACSArias^TM^ II flow cytometry machine. After few generations of cell passaging process, lentiviral encoding vector control, SIRT3 WT, and SIRT3 K288R stable transfectants were obtained with GFP^+^ cells greater than 90%.

### 4.5. Ubiquitination Assay

A total of 10 million 293 T cells were co-transfected with FLAG-tagged ubiquitin plasmid and lentiviral encoding empty vector plasmid for 24 h followed by momordin-Ic for additional 24 h. Cells were then incubated with MG132 for 4 h and lysed in radioimmunoprecipitation assay (RIPA) buffer (50 mM Tris [pH 7.4] containing 1% NP-40, 0.1% SDS, 1% sodium deoxycholate, 150 mM NaCl, 5 mM EDTA, 50 mM NaF, and 1mM phenylmethylsulfonyl fluoride [PMSF]), immunoprecipitated with the anti-SIRT3 antibody overnight at 4 °C, and subsequently incubated with Protein A/G Magnetic Beads (MCE, North Brunswick Township, NJ, USA) for 4 h. The precipitated proteins were washed three times with RIPA buffer without sodium deoxycholate and SDS, resolved by SDS PAGE, and visualized after Western blot using the anti-FLAG antibody (Sigma-Aldrich, St. Louis, MO, USA).

### 4.6. Co-Immunoprecipitation and Immunoblotting

AML cells were lysed with lysis buffer [1M Tris-HCl (pH 6.8 at 25 °C), 10% SDS, 50% glycerol, 0.05% bromophenol blue, and 10% β-mercaptoethanol] prior to incubating on ice for 30 min. Total protein lysates were harvested by centrifugation at 12,000× *g* for 10 min at 4 °C. Co-IP magnetic beads that purchased from MCE (Seattle, WA, USA) were prepared per manufacturer’s instruction prior to incubation with appropriated antibodies for 2 h, followed by an additional 2 h incubation with protein lysates at RT. The magnetic beads were then eluted with elution buffer containing loading dye before electrophoresed on 10% acrylamide gel and finally transferred onto PVDF (GE Healthcare, Chicago, WA, USA) membrane. The membrane was then blocked-in tris-buffered saline–Tween [Tris 50 mM (pH 8), NaCl 15 mM, 0.1% Tween] supplemented with 5% milk, followed by incubating with appropriate primary antibodies for overnight at 4 °C. Membranes were washed and probed either with horseradish peroxidase-conjugated goat anti-rabbit or anti-mouse secondary antibodies.

### 4.7. Annexin V/7AAD Apoptosis Detection Assay

Detection of apoptosis was performed using the Annexin V-APC Apoptosis *Detection* Kit II (BD Bioscience, San Jose, CA, USA) according to manufacturer’s protocol.

### 4.8. Metabolism Assays

NADP/NADPH (Abcam, London, UK), GSH/GSSG (Abcam, London, UK), ATP (Beyotime, Shanghai, China), JC-1(Beyotime, Shanghai, China), FAO (Abcam, London, UK) assays were performed according to manufacturers’ instructions.

### 4.9. Extracellular Acidification Rate and Basal Oxygen Consumption Rate

Oxygen consumption rates (OCR) and extracellular acidification rates (ECAR) assays were performed as per instruction manual (Seahorse Bioscience, North Billerica, CA, USA). Briefly, AML cells were treated with either 2.5 μM Ara-C or vehicle control for 48 hrs. Cells were then seeded in duplicates at the density of 5 × 10^5^ in an XF96 cell culture microplate, which pre-coated with Corning^®^ Cell-Tak™ Cell and Tissue Adhesive (Corning Incorporated, Somerville, NY, USA) to allow adhesion of suspension cells. To test mitochondria respiration, sequential compound injections, including 1 mM oligomycin A, 1 mM carbonyl-cyanide p-trifluoromethoxyphenylhydrazone (FCCP), 0.5 mM antimycin A/rotenone, were applied on the microplate after analyzer calibration. To test glycolytic activity, 100 mM glucose, 100 mM oligomycin A and 500 mM 2-DG, were sequentially injected into the microplate followed by calibration step. Data were analyzed by Wave 2.2.0 software (Santa Clara, CA, USA).

### 4.10. Transcriptome Sequencing (RNA-Seq) and Data Processing

The mRNA was isolated from total RNA using NEB Next poly (A) mRNA Magnetic Isolation Module (NEB, Ipswich, MA, USA) and rRNA was removed using RiboZero Magnetic Gold Kit (Illumina, CA, USA). RNA library was prepared using KAPA Stranded RNA-SEQ Library Prep Kit (Illumina, CA, USA) according to the instruction manual prior to subject to sequence using Illumina NovaSeq 6000. RNA library was qualified by Agilent 2100 Bioanalyzer and quantified by qPCR analysis. Raw sequencing data were QC qualified and trimmed data were aligned with reference genome/transcriptome (GRCh37). Differentially expressed genes or transcripts were subjected to either pathway or GO analysis. Venn graph and heatmap were generated using R language.

### 4.11. Drug Synergy

AML cells were seeded into 96-well plates at approximately 10,000 cells/well and allowed to grow for 24 h. Ara-C (2.5, 5, 10, 15 or 20 μM), momordin-Ic (10, 15, 20, 25 or 30 μM), or a combination of these two drugs were added to each well in triplicates. Cells were then stained with 7AAD for viability assay. Cell viability results were normalized to vehicle controls and then inputted into the CalcuSyn program (http://www.biosoft.com/w/calcusyn.htm) that calculated each combination index (CI) value using the Chou and Talalay method, where CI < 1 indicates synergy, CI = 1 indicates additive effect and CI > 1 indicates antagonism.

### 4.12. Animal Studies

To investigate the effect of SIRT3 activity on tumorigenesis in vivo, xenotransplants were established by i.v. injecting 5 × 10^6^ of lentiviral encoding vector control, SIRT3 or SIRT3K288R transduced AML cells per mouse into sub-lethally irradiated with 2.5 Gy mice. Human AML engraftment was determined at 4–6 weeks post-transplantation. To further explore synergized effects of Ara-C and momordin-Ic in vivo, xenotransplants were established as mentioned previously. Approximately 4 weeks post-transplantation, mice were treated with either PBS, Ara-C (45 mg/kg) for 5 consecutive days, momordin-Ic (10 mg/kg) every two days for a week or combination of Ara-C and momordin-Ic. Animals were sacrificed, tumor burden and mice survival were monitored.

### 4.13. Statistical Analyses

Two-sided student’s t-test was used to compare differences between two groups of cells in vitro. Two-way ANOVA was used to compare the differences between more than two groups. Data were plotted using GraphPad Prism 8.0 software and presented as means ± SD.

## 5. Conclusions

In summary, our data revealed that chemotherapy induces SIRT3 de-SUMOylation, which confers AML chemoresistance possibly through down-regulation of HES1-dependent FAO and targeting of SIRT3 de-SUMOylation synergize with chemotherapeutic agent Ara-C in vitro and in vivo, may be a promising regimen to overcome chemoresistance and improve the clinical outcome in AML (Appendix A).

## Figures and Tables

**Figure 1 ijms-23-08282-f001:**
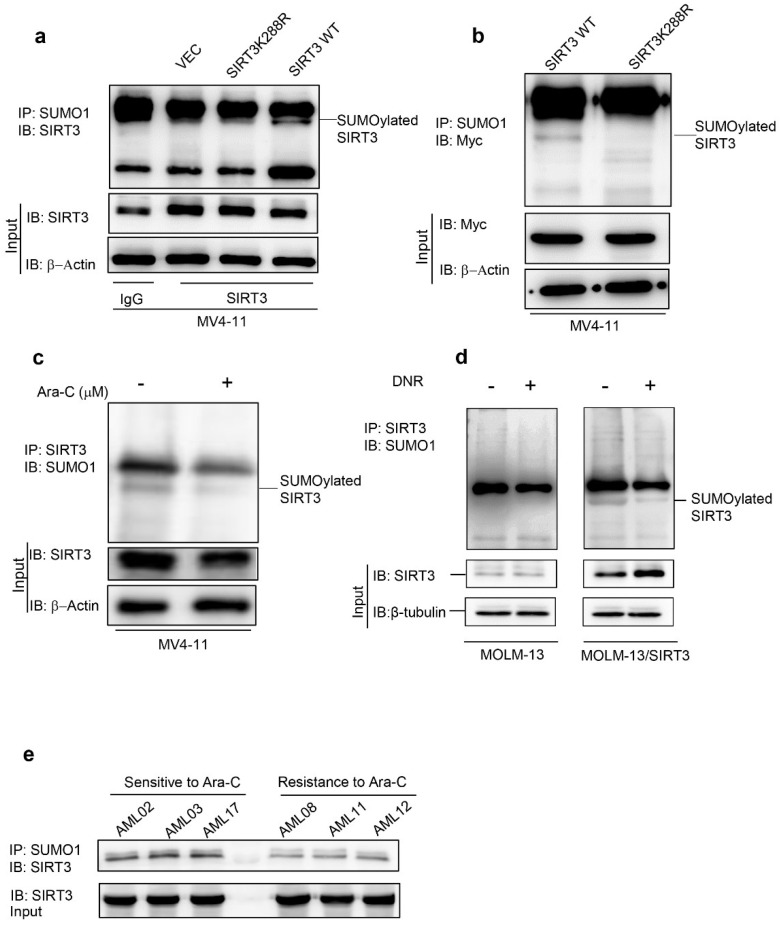
Chemotherapy induces SIRT3 de-SUMOylation in AML. Whole cell lysates were isolated from lentiviral encoding vector control, SIRT3 and SIRT3K288R overexpressing MV4-11 cells before immunoprecipitated with anti-SUMO1 antibody followed by Immunoblotting with either (**a**) anti-SIRT3 antibody or (**b**) anti-myc antibody. Whole cell lysates were isolated from SIRT3 WT overexpressing MV4-11 cells treated either with or without 2.5 mM Ara-C (**c**) or 50 nM DNR (**d**) for 48 h prior to subject to immunoprecipitation with anti-SIRT3 antibody, followed by immunoblotting with anti-SUMO1 antibody. (**e**) Whole cell lysates were isolated from three de novo and three relapse primary AML cells prior to subject to immunoprecipitation with anti-SUMO1 antibody, followed by Western blot with anti-SIRT3 antibody. Rabbit IgG was used as a negative control. The 10% input proteins were loaded to determine the expressions of endogenous SIRT3, and b-Actin was included as an indication of equal loading. Data in (**a**–**e**) are representative of the mean +SD from technical triplicates.

**Figure 2 ijms-23-08282-f002:**
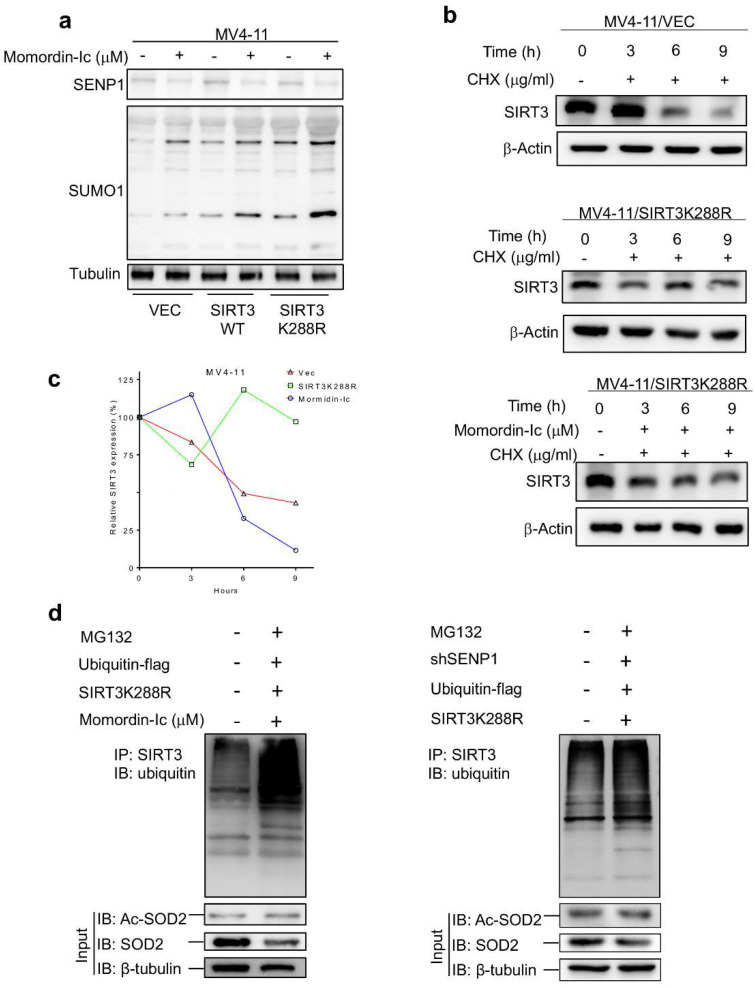
De-SUMOylation activates SIRT3 via inhibition of its protein degradation. (**a**) Immunoblotting assay of SIRT3 protein expression in aforementioned cells. (**b**) Densitometry quantification of SIRT3 protein (normalized to 0 h time point) was represented in the left panel. MV4-11 cells were co-transduced with FLAG tagged ubiquitin plasmid and lentiviral encoding SIRT3K288R plasmid prior to either (**c**) treated with 20 mM momordin-Ic or (**d**) co-transduced with shSENP1 for 48 h followed by 20 mM of MG132 for additional 4 h. Cell lysates were harvested, and equal amounts of proteins were subjected to immunoprecipitation assay. SIRT3 was immunoprecipitated with anti-SIRT3 antibody followed by immunoblotting with anti-FLAG antibody. A 10% input protein was loaded to determine the expressions of endogenous SIRT3, acetylated SOD2 and total SOD2 proteins. Either b-Actin or tubulin was included as an indication of equal loading. Data in (**a**–**d**) are representative of the mean +SD from technical triplicates.

**Figure 3 ijms-23-08282-f003:**
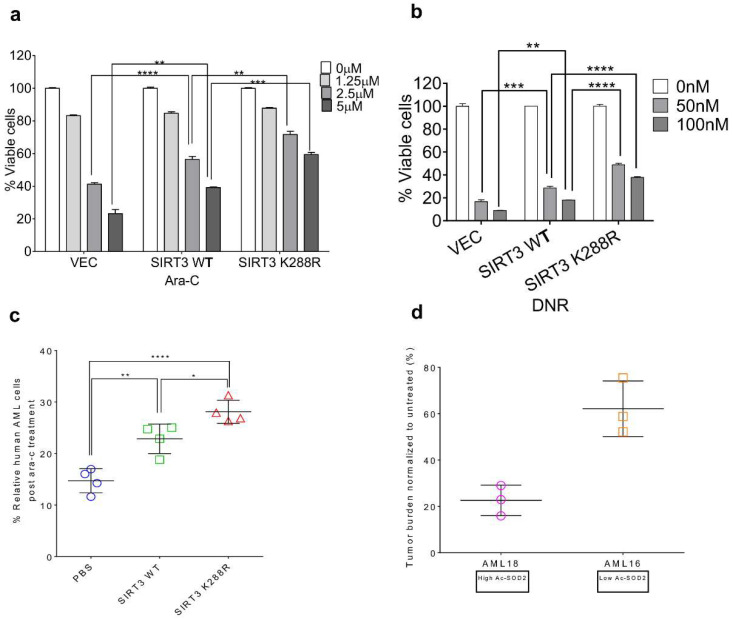
SIRT3 de-SUMOylation contributes to AML chemoresistance in vitro and in vivo. MV4-11 cells transduced with vector control, SIRT3, and SIRT3K288R were treated with indicated doses of (**a**) Ara-C or (**b**) DNR for 48 h. Cell viability was analyzed by flow cytometry upon treatment. (**c**) 6–8-week-old female NOD/SCID mice were irradiated with half lethal dose of 2.5 Gy prior to subject to xenotransplantation either with lentiviral encoding vector control, SIRT3, and SIRT3K288R overexpressing MV4-11 cells via tail vein injection. Four weeks post-transplantation, mice were i.p injected with Ara-C for 5 consecutive days and then sacrificed for AML engraftment analysis by flow cytometry. (**d**) Primary cells were isolated from PDX AML xenografts that derived from AML16 and AML18 as described previously [6], and cells were then treated with 2.5 mM Ara-C for 48 h prior to subject to flow cytometry for tumor burden (% of AML engraftment relative to untreated). Data in (**a**–**d**) are representative of the mean +SD from technical triplicates. (* *p* < 0.05; ** *p* < 0.01; *** *p* < 0.001; **** *p* < 0.0001).

**Figure 4 ijms-23-08282-f004:**
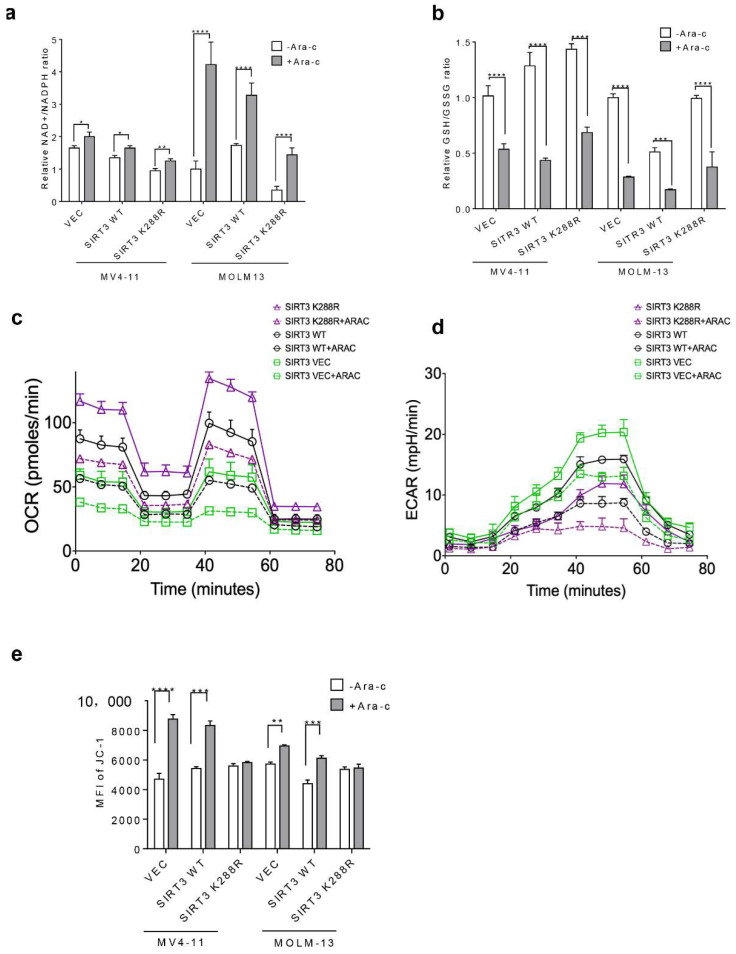
SIRT3 SUMOylation regulates mitochondria biogenesis in AML cells (**a**) NADP/NADPH and (**b**) GSH/GSSG ratios were analyzed by the corresponding kits and absorbance. MV4-11 cells transduced with vector control, SIRT3, or SIRT3K288R were treated either with (dash lines) or without (solid lines) 2.5 mM of Ara-C for 48 h. Cells were then seeded at the density of 1 × 10^6^/50 mL in a Cell Tak coated FX 96 well plate and washed with base medium. (**c**) OCR and (**d**) ECAR were determined by Seahorse Agilent. (**e**) MMP was analyzed by JC-1 staining in MV4-11 and MOLM-13 stable transfectants treated with Ara-C for 48 h. Data in (**a**–**e**) are representative of the mean +SD from technical triplicates. (* *p* < 0.05; ** *p* < 0.01; *** *p* < 0.001; **** *p* < 0.0001).

**Figure 5 ijms-23-08282-f005:**
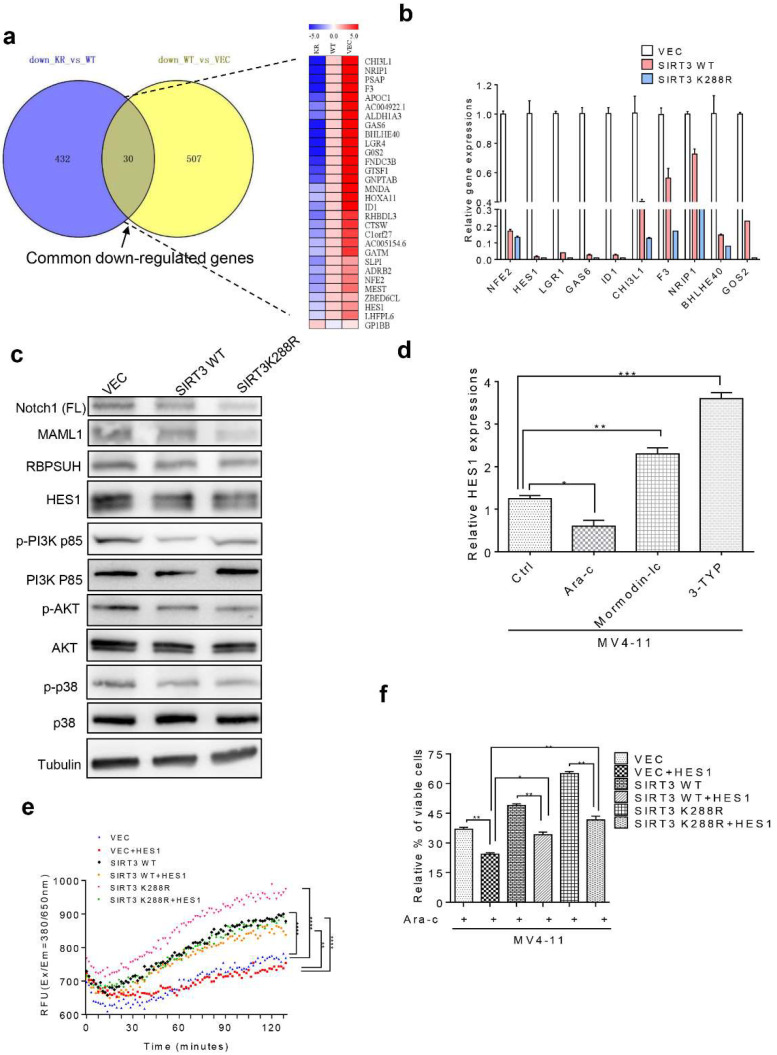
SIRT3 SUMOylation modulates AML chemoresistance via downregulating of HES1 dependent FAO. (**a**) MV4-11 cells transduced either with vector control, SIRT3 or SIRT3K288R were subjected for RNA-Seq analysis, and 30 simultaneously downregulated genes were selected based on their fold change and PFKM value. (**b**) Real-time PCR validation of the most significant down-regulated genes in MV4-11 stable transfectants. (**c**) Immunoblotting analysis of expressions of notch1, MALM1, BPUSH, HES1, phosphorylated and total of PI3K, ATK, and p38 proteins. Tubulin has included an indication of equal loading. (**d**) HES1 mRNA level was determined by qRT-PCR assay in MV4-11 cells treated with Ara-C, momordin-Ic, or 3-TYP and was represented as relative expression normalized to vehicle control. (**e**) FAO was determined in SIRT3K288 lentiviral plasmid overexpressing MV4-11 cells either treated with or without 20 mM Etomoxir. (**f**) Cell viability was assessed in SIRT3 K288R lentiviral plasmid overexpressing MV4-11 cells either treated with or without Ara-C, Etomoxir alone or both for 48 h. FAO or cell viability was assessed in MV4-11 cells transduced either with lentiviral encoding vector control, SIRT3, SIRT3K288R alone or co-transduced with lentiviral encoding HES1 treated with Ara-C for 48 h. RFU: relative fluorescence units. Each dot represents RFU per minute. Data in (**a**–**f**) are representative of the mean +SD from technical triplicates. (* *p* < 0.05; ** *p* < 0.01; *** *p* < 0.001).

**Figure 6 ijms-23-08282-f006:**
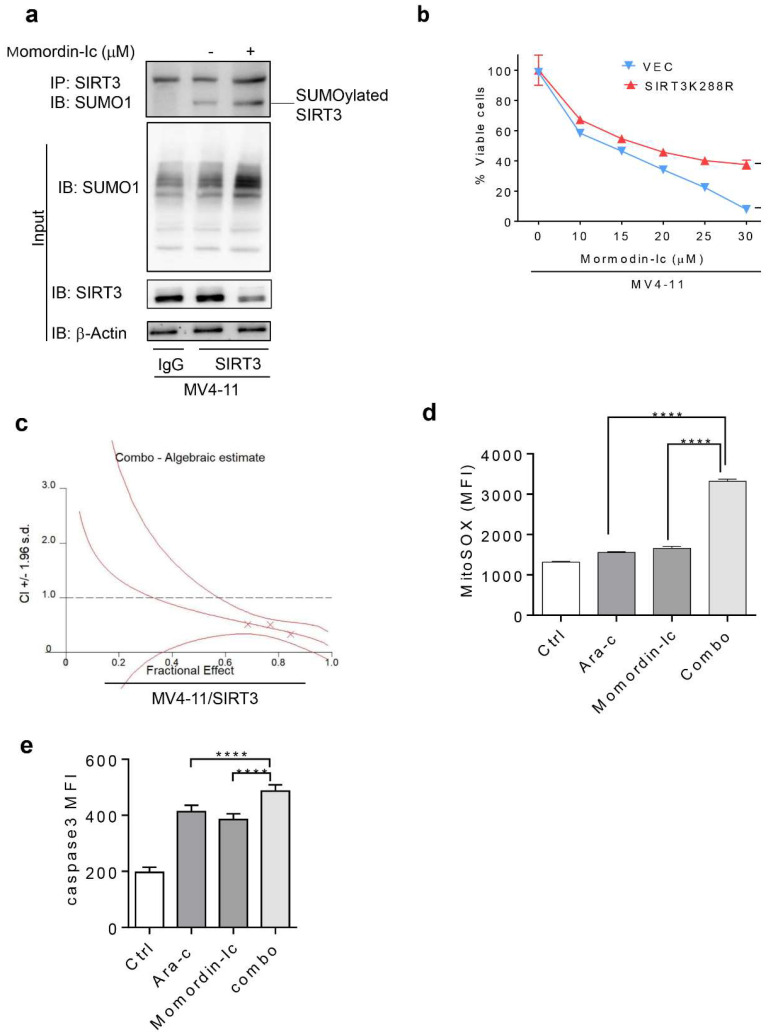
Inhibition of SIRT3 de-SUMOylation synergizes with Ara-C in AML in vitro. (**a**) MV4-11 cells were treated with 20 mM of momordin-Ic for 48 h. The whole cell protein lysates were extracted as previously described and immunoprecipitated with anti-SIRT3 antibody before probed with anti-SUMO1 antibody. A 10% input protein was loaded to determine the endogenous expression of SIRT3 protein. Rabbit IgG was included as a negative control. (**b**) MV4--11/VEC or MV411/SIRT3K288R cells were treated with momordin-Ic with various doses of 10, 15, 20, 25, and 30 mM for 48 h, and cell viability was determined by annexin V/7--AAD staining and flow cytometry analysis. (**c**) MV411/SIRT3K288R cells were treated with Ara--C (0.5, 1, 2, 5, and 10 mM) or momordin-Ic (10, 15, 20, 25, 30 mM) alone, or combined drugs with a constant ratio (Ara--C: momordin-Ic = 1:5) for 48 h. Cell viability was determined by annexin V/7--AAD staining and flow cytometry analysis. Synergistic effect between Ara--C and momordin-Ic was shown as CI < 1. (**d**) MV4-11 cells treated with Ara-C or momordin-Ic alone or combo were stained with MitoSOX and shown as overlaid histogram. (**e**) MV4-11 cells treated either with Ara-C, momordin-Ic alone, or combination for 48 h. Cells were then permeabilized and stained with Alexa Fluor 647-conjugated antibody against active caspase 3 followed by flow cytometry analysis. Data in (**a**–**e**) are representative of the mean +SD from technical triplicates (**** *p* < 0.0001).

**Figure 7 ijms-23-08282-f007:**
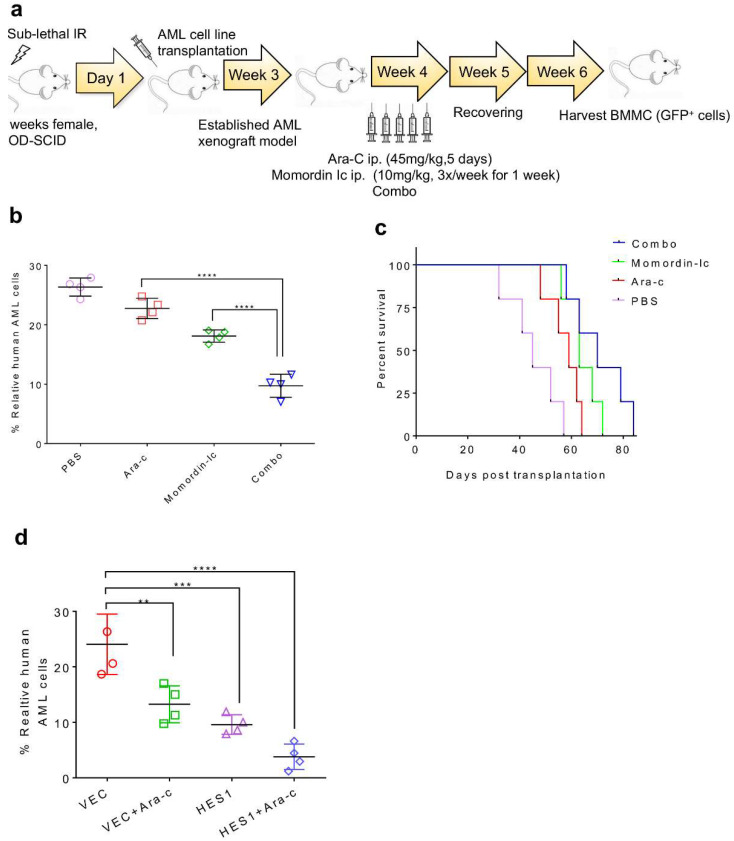
Combination therapy targeting SIRT3 SUMOylation pathways (**a**) schematic illustration of therapeutic regimen in AML xenografted mouse model. MV4-11engrafted NOD/SCID mice were treated either with Ara-C, momordin-Ic alone or combo at the indicated time and doses. A week after drug treatment, mice were sacrificed and (**b**) tumor burden, (**c**) survival curve was determined by percent of GFP+ cells in the murine bone marrow. (**d**) SIRT3 alone transduced or SIRT3 and HES1 co-transduced MV4-11 cells engrafted NOD/SCID mice were treated with Ara-C at the indicated interval and doses. A week after drug treatment, mice were sacrificed, and percentage of AML engraftment was assessed by flow cytometry analysis. Data in (**b**–**d**) are representative of the mean +SD from technical triplicates (** *p* < 0.01; *** *p* < 0.001. **** *p* < 0.0001).

## Data Availability

RNA-Seq data have been deposited in GEO database (accession No: GSE179617, Reviewer private assess token: ejkdoqmavxkvlkr). GEO link: https://www.ncbi.nlm.nih.gov/geo/info/linking.html assessed on 7 July 2021.

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
