# Peer review of "Dysregulation of SIRT3 SUMOylation Confers AML Chemoresistance via Controlling HES1-Dependent Fatty Acid Oxidation"

_ijms, 2022, doi:10.3390/ijms23158282_

Round 1

Reviewer 1 Report

Major point

This study describes a novel role of SIRT3 SUMOylation in regulation of chemoresistance in AML cells. In general, study is well designed and carried out. Experimental design and methods are appropriate. However, interpretation of results are arguable. Major point for the debate is the conclusion that SIRT-3 de-SUMOylation confers AML chemoresistance via down-regulating HES1-dependent fatty acid oxidation (FAO) process. Authors do not provide any evidence of inhibition of FAO. It is true that downregulation of HES1 enhances FAO, presumably via crosstalk between Notch and 173 PI3K/AKT signaling pathways. However, HES1 inhibition will have far wider effects on cellular and metabolic processes.  Authors showed that overexpression of mutant SIRT3 stimulated mitochondrial respiration and inhibited extracellular acidification rate. However, they didn’t show changes in fatty acid oxidation reactions. Authors should provide indisputable evidence that FAO fluxes are affected by SIRT3 SUMOylation / de-SUMOylation.

Other points

Figure 4 (c and d): additions of reagents in Seahorse measurements (FCCP, antimycin A, rotenone, etc.) and their concentrations should be marked.

Greek symbols are missing in all figure legends.

Lines 91-92 and Figure 2b: Are these gels represent cells transfected with empty vector, or encoding WT protein?

Lines 178-179: The sentence needs to be finished.

Lines 205-206: The sentence needs revision. "…were more resistant" than what?

Lines 277 and 284: Sentences should be revised, e.g. Increasing number of studies (or reports).

Figure S1. According to the legend, three lines are supposed to be shown. However, only two lines are present on this figure

Author Response

Reviewer #1:

Major point

This study describes a novel role of SIRT3 SUMOylation in regulation of chemoresistance in AML cells. In general, study is well designed and carried out. Experimental design and methods are appropriate. However, interpretation of results are arguable. Major point for the debate is the conclusion that SIRT-3 de-SUMOylation confers AML chemoresistance via down-regulating HES1-dependent fatty acid oxidation (FAO) process. Authors do not provide any evidence of inhibition of FAO. It is true that downregulation of HES1 enhances FAO, presumably via crosstalk between Notch and 173 PI3K/AKT signaling pathways. However, HES1 inhibition will have far wider effects on cellular and metabolic processes.  Authors showed that overexpression of mutant SIRT3 stimulated mitochondrial respiration and inhibited extracellular acidification rate. However, they didn’t show changes in fatty acid oxidation reactions. Authors should provide indisputable evidence that FAO fluxes are affected by SIRT3 SUMOylation / de-SUMOylation.

Response to major point: Author would like to thank reviewer’s valuable comments and suggestions. We agreed that Notch and PI3K/AKT signaling pathway cross talk may not be the only pathways that modulating HES1, however, several studies were demonstrated the importance of their interwind in leukemogenesis such as T cell acute lymphoblastic leukemia (T-ALL) (Cancer Cell. 2007 Nov;12(5):411-413; J Cell Biochem. 2008 Apr 1;103(5):1405-12; Int J Hematol Oncol Stem Cell Res. 2020 Apr 1;14(2):99-109.). To demonstrate the importance of FAO in the regulation of chemoresistance in AML cells, etomoxir, an FAO inhibitor, was used (Figure 5e). In fact, inhibition of FAO dampened SIRT3K288R mediated AML chemoresistance (Figure 5f). We have demonstrated that overexpression of HES1 argumented SIRT3K288R induced FAO production (Figure 5g). Moreover, overexpression of HES1 counteracted SIRT3K288R (mutant) induced AML chemo-resistance by decreasing AML cells survival (Figure 5h. Taken together, these results indicate that SIRT3 de-SUMOylation mediated AML chemoresistance may via HES1 dependent FAO in AML (lines 200-208, 226-229, 326-330). Indeed, in vivo approach such as HES1 antagonist need to be performed to further validate this finding. 

Other points

Figure 4 (c and d): additions of reagents in Seahorse measurements (FCCP, antimycin A, rotenone, etc.) and their concentrations should be marked.

Concentrations of above additional reagents were listed in the main text (lines 426-429).

Greek symbols are missing in all figure legends.

All Greek symbols were corrected as suggested by the reviewer (lines 81, 86, 125, 126, 130, 159, 178, 253, 257, 259 & 260).

Lines 91-92 and Figure 2b: Are these gels represent cells transfected with empty vector, or encoding WT protein?

Figure 2b top panel represents AML cells transfected with empty vector lentiviral plasmids, whereas middle and bottom panels illustrate AML cells transduced with SIRT3 de-SUMOylation lentiviral plasmids SIRT3K288R.

Lines 178-179: The sentence needs to be finished.

This sentence was completed as per reviewer’s request (lines 198-200).

Lines 205-206: The sentence needs revision. "…were more resistant" than what?

This sentence was modified as appropriates (lines 241-242).

Lines 277 and 284: Sentences should be revised, e.g. increasing number of studies (or reports).

This sentence was modified as per reviewer’s request (line 316).

Figure S1. According to the legend, three lines are supposed to be shown. However, only two lines are present on this figure

Figure legend of Figure S1 was modified as only two lanes were applied in this experiment.

Reviewer 2 Report

1.     In the Abstract section, the authors used a lot of abbreviations and it is suggested to provide full spellings when abbreviations first appear, such as SIRT3, AML, SENP1, HES1 and FAO.

2.     In the Introduction section, the background is not sufficiently introduced. It is suggested to add more research background introduction. For example, there are seven members of sirtuin family, why only SIRT3 is selected, and what functions the other members play.

3.     In the Materials and Methods section, it is suggested to add the company and origin of each reagent used in your experiments, such as IMDM, FBS and penicillin/streptomycin (from line 317 to 319).

4.     There is a lot of extra Spaces in the manuscript and it is suggested to revise them, such as in line 69, 74, 109, 110, 114, 141, 158, 221, 223, 224, and 410.

5.     As for reference insertion, there is Spaces or no Spaces between the reference number and the previous words; please keep them the same, such as in line 285, and some other places.

6.     As for “in vitro” and “in vivo”, some places use italics and some places use formal; please be consistent, such as in line 20, 117, 120, 405, 409, 423, etc.

7.     There are some grammatical errors in the manuscript. It is suggested that the article be revised and polished by professional.

Author Response

Reviewer #2:

Author would like to thank reviewer’s valuable comments and suggestions.

  1. In the Abstract section, the authors used a lot of abbreviations and it is suggested to provide full spellings when abbreviations first appear, such as SIRT3, AML, SENP1, HES1 and FAO.

Response to point 1: Full names were given to all abbreviations (lines 11, 12, 13, 19 & 20)

  1. In the Introduction section, the background is not sufficiently introduced. It is suggested to add more research background introduction. For example, there are seven members of sirtuin family, why only SIRT3 is selected, and what functions the other members play.

Response to point 2: A paragraph of Sirtuin family proteins was added accordingly (lines 30-38).

  1. In the Materials and Methods section, it is suggested to add the company and origin of each reagent used in your experiments, such as IMDM, FBS and penicillin/streptomycin (from line 317 to 319).

Response to point 3: Vendor information and Cat. No. were added in the Materials and Methods section (lines 363-366).

  1. There is a lot of extra Spaces in the manuscript and it is suggested to revise them, such as in line 69, 74, 109, 110, 114, 141, 158, 221, 223, 224, and 410.

Response to point 4: Author would like to thank reviewer to point this issue out, however, we only be able to find one extra space in line 410 which has been corrected.

  1. As for reference insertion, there is Spaces or no Spaces between the reference number and the previous words; please keep them the same, such as in line 285, and some other places.

Response to point 5: Spaces between the reference and the previous words were removed for all in-text citations.

  1. As for “in vitro” and “in vivo”, some places use italics and some places use formal; please be consistent, such as in line 20, 117, 120, 405, 409, 423, etc.

Response to point 6: Author replaced formal format of ‘in vivo’ (lines 96, 133, 137, 152, 351, 453, 457, 471) and ‘in vitro’ with italics (lines 133, 134, 152, 235, 252, 464, 471).

  1. There are some grammatical errors in the manuscript. It is suggested that the article be revised and polished by professional

Response to point 7: MS has already polished by one of my colleagues.

Round 2

Reviewer 1 Report

Authors addressed my concerns. Revised manuscript can be published in present form.

Reviewer 2 Report

The manuscript has been carefully revised and is recommended for acceptance.